# Viral Population Diversity during Co-Infection of Foot-And-Mouth Disease Virus Serotypes SAT1 and SAT2 in African Buffalo in Kenya

**DOI:** 10.3390/v14050897

**Published:** 2022-04-25

**Authors:** Rachel M. Palinski, Barbara Brito, Frederick R. Jaya, Abraham Sangula, Francis Gakuya, Miranda R. Bertram, Steven J. Pauszek, Ethan J. Hartwig, George R. Smoliga, Vincent Obanda, George P. Omondi, Kimberly VanderWaal, Jonathan Arzt

**Affiliations:** 1Foreign Animal Disease Research Unit, Plum Island Animal Disease Center, ARS USDA, Orient Point, NY 11957, USA; rpalinski@vet.k-state.edu (R.M.P.); miranda.bertram@usda.gov (M.R.B.); ethan.hartwig@usda.gov (E.J.H.); george.smoliga@usda.gov (G.R.S.); 2The Australian Institute for Microbiology & Infection, University of Technology Sydney, Ultimo 2007, Australia; barbara.britorodriguez@uts.edu.au (B.B.); frederick.r.jaya@student.uts.edu.au (F.R.J.); 3Kenya Foot-and-Mouth Disease Laboratory, Embakasi, Nairobi 00500, Kenya; aksangula@gmail.com; 4Veterinary Science and Laboratories Department, Wildlife Research and Training Institute, Naivasha 20117, Kenya; gakuya@kws.go.ke (F.G.); vobanda@gmail.com (V.O.); 5Foreign Animal Disease Diagnostic Laboratory, National Veterinary Services Laboratories, Animal and Plant Health Inspection Service, United States Department of Agriculture, Plum Island Animal Disease Center, Orient Point, NY 11957, USA; steve.pauszek@usda.gov; 6Department of Veterinary Population Medicine, College of Veterinary Medicine, University of Minnesota, St. Paul, MN 55108, USA; paulx176@umn.edu; 7Ahadi Veterinary Resource Center, Nairobi 20117, Kenya

**Keywords:** foot-and-mouth disease, short-read sequencing, intra-host diversity, serotype A

## Abstract

African buffalo are the natural reservoirs of the SAT serotypes of foot-and-mouth disease virus (FMDV) in sub-Saharan Africa. Most buffalo are exposed to multiple FMDV serotypes early in life, and a proportion of them become persistently infected carriers. Understanding the genetic diversity and evolution of FMDV in carrier animals is critical to elucidate how FMDV persists in buffalo populations. In this study, we obtained oropharyngeal (OPF) fluid from naturally infected African buffalo, and characterized the genetic diversity of FMDV. Out of 54 FMDV-positive OPF, 5 were co-infected with SAT1 and SAT2 serotypes. From the five co-infected buffalo, we obtained eighty-nine plaque-purified isolates. Isolates obtained directly from OPF and plaque purification were sequenced using next-generation sequencing (NGS). Phylogenetic analyses of the sequences obtained from recombination-free protein-coding regions revealed a discrepancy in the topology of capsid proteins and non-structural proteins. Despite the high divergence in the capsid phylogeny between SAT1 and SAT2 serotypes, viruses from different serotypes that were collected from the same host had a high genetic similarity in non-structural protein-coding regions P2 and P3, suggesting interserotypic recombination. In two of the SAT1 and SAT2 co-infected buffalo identified at the first passage of viral isolation, the plaque-derived SAT2 genomes were distinctly grouped in two different genotypes. These genotypes were not initially detected with the NGS from the first passage (non-purified) virus isolation sample. In one animal with two SAT2 haplotypes, one plaque-derived chimeric sequence was found. These findings demonstrate within-host evolution through recombination and point mutation contributing to broad viral diversity in the wildlife reservoir. These mechanisms may be critical to FMDV persistence at the individual animal and population levels, and may contribute to the emergence of new viruses that have the ability to spill-over to livestock and other wildlife species.

## 1. Introduction

Foot-and-mouth disease (FMD) is one of the most economically significant transboundary diseases affecting livestock and wild ungulates. The etiologic agent of FMD, foot-and-mouth disease virus (FMDV), is a highly transmissible picornavirus that causes fever and characteristic vesicular lesions on the feet and oral cavities in clinically infected animals [1,2]. The seven distinct FMDV serotypes (A, Asia1, C, O, SAT 1–3) are further classified into topotypes, lineages, and sublineages based on the VP1 coding sequence [3]. The Southern African Territories (SAT) serotypes have been historically endemic and limited to sub-Saharan Africa, but have caused outbreaks in North Africa and the Middle East [4,5]. FMDV SAT serotypes’ natural reservoir is the African (Cape) buffalo (*Syncerus caffer*), though SAT infection can also occur in other wildlife species, and can circulate in domestic livestock in Africa [6,7].

In domestic livestock, FMD is characterized by an acute clinical phase of infection consisting of fever and vesicles that resolve within two weeks. However, FMDV also causes long-term subclinical, persistent infection in epithelial cells of the nasopharynx of cattle [8,9] and oropharyngeal tonsils of sheep [10,11]. Additionally, there is an early form of subclinical infection (neoteric phase) which has higher levels of shedding and transmissibility compared to the persistent phase [9]. In African buffalo, acute clinical disease is not frequent [7], and persistent infection occurs in the palatine tonsil [12].

Although there is high FMDV seroprevalence in some buffalo populations [13], and contact between African buffalo and domestic cattle herds occurs in shared grazing areas, the frequency of FMDV spill-over from buffalo to livestock is not well understood [14,15]. Molecular (genomic) epidemiology can provide a framework to elucidate disease transmission by analyzing the genetic similarity between viruses sampled from buffalo and cattle [15,16,17]. In Kenya, the only two published whole genomes of buffalo-derived FMDV sequences (one SAT1 and one SAT2) are genetically distinct from those isolated from cattle, suggesting a low frequency of transmission across species [18]. However, the lack of available reference genomes prevents explicit understanding of the relationship between livestock- and buffalo-derived viruses.

Point mutation of RNA viruses during fast-occurring viral replication and absence of proof-reading ability of the RNA polymerase results in a diversity of viral genomes existing within one host [19]. Additionally, African buffalo can sustain multi-serotype FMDV-persistent infections simultaneously, providing conditions for related viruses to recombine [12,20]. All of these evolutionary mechanisms may contribute to alterations of viral fitness, host range, and transmissibility.

The current NGS technologies based on short sequencing reads have provided a platform to efficiently perform whole viral genome sequencing. However, technical challenges may arise when bioinformatically assembling the fragmented nucleic acid into representative consensus viral haplotypes. These challenges are exacerbated by the presence of multiple related viruses in co-infected animals. Under such circumstances, techniques that enable the isolation of each virus separately may be required to obtain the multiple viral genomes present in one sample.

In this study, we acquired complete FMDV genome sequences from African buffalo naturally infected with FMDV in Kenya. We further characterized the within-host FMDV genetic diversity in animals co-infected with multiple serotypes. This improved understanding of viral evolution during co-infection contributes to elucidation of the emergence of new variants, and may enable the prediction of alterations of host range and transmission of FMDV.

## 2. Materials and Methods

### 2.1. African Buffalo Sample Collection

Samples were collected from African buffalo located within the Ol Pejeta Conservancy (OPC) in central Kenya during January 2016 (Figure 1). Within and beyond the bounds of this conservancy, approximately 1200 buffalo travel in herds of 30–200 individuals. The fence surrounding the conservancy freely allows animal movement to maintain contiguity of the larger Laikipia–Samburu–Isiolo ecosystem. In addition to wildlife conservation efforts, OPC maintains ~8500 cattle which are allowed to graze and intermingle with wildlife during the day but are corralled at night. The Ewaso Ngiro River forms the only natural boundary traversing the conservancy, in many cases preventing east–west animal movement across the river, and shaping the distribution of bacterial and viral pathogens (Figure 1) [15,21]. Cattle are vaccinated against FMDV at <12 months of age, but no FMDV control is performed on the buffalo. The fifty-four buffalo oropharyngeal fluid (OPF) samples utilized in this study were collected as described previously [15,22]. Briefly, Kenya Wildlife Service veterinarians anesthetized the buffalo with etorphine hydrochloride and azaperone via a dart gun. OPF samples were collected using a probang cup [22]. OPF samples were transported to Plum Island Animal Disease Center, Plum Island, NY, and stored at −80 °C until use.

In addition, two samples previously collected during an outbreak in cattle (VE; vesicular epithelium) in OPC in 2014 were sequenced and included in the phylogenetic analyses.

### 2.2. Virus Isolation

OPF and VE samples were filtered by centrifugation in Spin-X filter columns (0.45 uM, Sigma-Aldrich, Saint Louis, MO, USA) to remove contamination. The filtrate was used to infect LFBK-αvβ6 cells [23]. Virus Isolation (VI) was performed as previously described, and FMDV was confirmed by qRT-PCR on first-passage supernatants (‘first passage virus’) [24]. VI supernatants from all samples were subsequently subjected to random-and-targeted deep sequencing (described below).

### 2.3. Identification of Co-Infected First-Passage Buffalo Samples

FMDV-positive first-passage supernatant RNA was subjected to RT-PCR targeting the P1 FMDV genomic region as previously described [25]. Amplicons were assessed by agarose gel, and bands of the anticipated sizes were extracted and purified. Libraries were generated from gel purified amplicons using the Nextera XT library preparation kit (Illumina, San Diego, CA, USA), and sequenced on an Illumina NextSeq platform. Co-infected samples were identified in CLC Genomics Workbench v.10.0 by analyzing the VP1 sequence data. Co-infections were identified when reads mapped to references belonging to two different serotypes’ reference sequences. This procedure was applied to the entire cohort of 54 buffalo OPF samples [15], of which five were found to be co-infected with SAT1 and SAT2, whereas the remainder were infected with a single serotype (SAT1 *n* = 16, SAT2 *n* = 33, SAT1/2 *n* = 5) (Table 1).

### 2.4. Plaque Purifications

Co-infected samples were subjected to plaque purification to further characterize the viral genomes (Table 1). LFBK-αvβ6 cells were grown in 6-well plates to 90% confluence, then inoculated with serial dilutions (10^−1^, 10^−2^, 10^−3^, 10^−4^, 10^−5^, and 10^−6^) of co-infected VI supernatants in Calcium/Magnesium-supplemented PBS plus 1% calf-serum. One hour after infection at 37 °C, the inoculum was removed, and each well was overlaid with minimal essential media (Invitrogen, Waltham, MA, USA) containing 1.25% agarose. Twenty-four hours post-infection, approximately 20 plaques were picked for each sample, and individual plaques were used to infect LFBK-αvβ6 cells in separate wells of 90% confluent 24-well plates. Supernatants were collected the following day and stored at −80 °C until deep sequence analysis.

### 2.5. Deep Sequencing of Virus Isolates and Plaques

Total RNA was extracted from cell supernatants using the MagMAX 96 Total RNA isolation kit (Ambion, Austin, TX, USA) as specified by the manufacturer. Subsequently, the total cell supernatant RNA was subjected to a random-and-targeted viral deep sequencing preparation similar to previously described methods with slight modifications [26]. Briefly, remaining host DNA was depleted using the DNA-free DNase kit (Ambion) per the manufacturer’s protocol. Treated RNA underwent first strand synthesis using the Superscript II first–strand synthesis system (Invitrogen) using random hexamer primers alone or random hexamer primers in combination with a 2A-specific (GCCCRGGGTTGGACTC) and a tagged oligo(dT) (ACGCTCGACATTTTTTTTTTTTTTTT) primer. Subsequently, the NEBnext (Ipswich, MA, USA) ultra non-directional RNA second strand synthesis module was utilized to generate double-stranded cDNA, which was purified using SPRI-select beads. Purified cDNA was quantified with the Qubit dsDNA HS Assay Kit (Thermo Fisher Scientific, Waltham, MA, USA) as specified by the manufacturer, and prepped for deep sequencing with the Nextera XT library preparation kit. Sequencing was performed on a NextSeq 500/550 platform using NextSeq Reagent kit v2 (Illumina). We performed sequencing on the first-passage isolates, as well as the plaque-purified isolates from the selected dual serotype infected samples.

### 2.6. Deep Sequence Data Analysis

A total of 112 plaque-purified isolates (Table 1), the 54 OPF isolates from the first passage, and the 2 outbreak isolates were deep-sequenced in duplicate, and full polyprotein open reading frames (ORFs) were assembled by either de novo or reference assembly methods (when the complete ORF was not obtained by the de novo assembly). The duplicate sequences for each sample were analyzed for similarity and quality control metrics. Sequences were assessed for completeness, similarity between duplicates, and SNP presence (first-passage (non-plaque-purified) samples only). Sequences were used in the analyses only if they fulfilled three criteria: (1) covered the complete polyprotein, (2) had 100% similarity between duplicate consensus sequences, and (3) had 100% identity to the previously published VP1 sequence (first-passage samples only) [17]. A total of 89 plaque sequences, as well as the 54 first-passage sequences and 2 outbreak sequences, met the criteria and were included in further analyses. Consensus sequences, reference mapping, and low-frequency variants (first-passage samples only) were generated from deep sequence data in CLC Genomics Workbench v11.0 using default parameters. Genomes were aligned and SNPs were visualized in Geneious Prime version 2019.2.3.

### 2.7. Assessment of Assembly of Co-Infected Samples

The consensus sequences of SAT1 and SAT2 assembled from the first passage (non-plaque purified) samples from co-infected animals (#6, #36, #51, #59, #61) were compared to the sequences obtained from each individual plaque to identify artifacts in the short read assembly process. We constructed phylogenies using the recombination-free areas using IQ-TREEv2.0 with the automatic selection of the substitution model (-m TEST) option [27]. Potential sequence artifacts would show in the phylogenetic tree as viruses distant from the plaque-purified groups.

### 2.8. Recombination Detection

A total of 149 sequences were included in recombination detection analyses. These corresponded to the 89 consensus sequences obtained from plaque-purified isolates from co-infected animals #51 (*n* = 9 SAT 1 and *n* = 9 SAT 2), #6 (*n* = 12 SAT 1 and *n* = 3 SAT 2), #36 (*n* = 7 SAT1, *n* = 12 SAT2), #59 (*n* = 21 SAT2), and #61 (*n* = 16 SAT2). The remaining sequences were the 54 first-passage sequences [16,25], 2 sequences from a cattle outbreak in 2014, and 4 closely related reference sequences published in GenBank: KEN_004/2002 (SAT 1, JF749860.1), TAN/22/2012 (SAT 1, KM268899.1), KEN_002/2002 (SAT 2JF749861.1), TAN/5/2012 (SAT 2KM268900.1). All sequences were aligned using MUSCLEv5 [28]. Recombination was detected in the alignment using RDP4 using default parameters [29]. The resulting breakpoint recombination plot (using permutations = 1000 and window size = 200 nucleotides) was used to determine the recombination breakpoints.

### 2.9. Time Divergence Estimation

The RDP4-detected breakpoints were used to partition the genomes into recombination-free segments. Plaque-purified sequences from co-infected samples, first-passage sequences from singly infected samples, and references as described in the recombination analysis methods were included in the analysis. Each segment was aligned, and a time divergence phylogeny was estimated using BEAST v1.8.4 [30]. A general time reversible model with a gamma distribution and an uncorrelated relaxed lognormal clock were used. A Bayesian skyline and an exponential growth tree prior were run for each of the segments. The best tree prior was selected based on stepping-stone marginal likelihood estimation [31]. The chain length for tree sampling was set to 500,000,000 trees and sampled every 50,000 iterations. The final maximum clade credibility (MCC) tree was annotated using FigTree v1.4.3. The 95% high posterior densities (95% HPD) for common ancestors were extracted from the MCC tree.

### 2.10. Within-Host Recombination Network Analysis

To visualize the within-host plaque virus diversity and haplotype relationships, a network analysis was performed in PopArt v1.7 [32] on potential chimeric viruses using a median joining algorithm [33].

### 2.11. Within-Animal Sequence Diversity

The set of sequences obtained from plaque assays from each ‘genotype’ identified in the phylogenetic analyses were analyzed separately to investigate within-animal sequence diversity. A genotype was defined as sequences that formed a monophyletic group across all protein-coding regions. The location and type of within-host nucleotide substitution based on the specific nucleotide change, transition or transversion, and synonymous or non-synonymous changes were identified in AliViewv3 [34].

## 3. Results

### 3.1. FMDV Sequence Acquisition

A total of 54 first-passage buffalo samples and 2 isolates from outbreak samples yielded complete polyprotein coding sequences (Appendix A). Five buffalo samples (ID = #6, #36, #51, #59, #61) were determined to be co-infected with FMDV-SAT1 and SAT2. These five samples underwent plaque purification, and complete polyprotein coding sequence was obtained from a total of 89/112 plaques (Table 1). Despite the identification of both SAT1 and SAT2 in the first-passage supernatants, viruses from animals #59 and #61 were all identified to be FMDV SAT2.

All SAT1 sequences obtained in this study (derived from OPF single and co-infected samples) belonged to the topotype I (NWZ), and SAT2 viruses belonged to topotype IV (Appendix A).

### 3.2. Evaluation of the Assembled Genomes from Original Co-Infected Samples Compared to the Plaque Sequences

To investigate the effects of bioinformatic tools on the assembly of genomes in co-infected samples (first-passage isolates), maximum likelihood trees were constructed for the SAT1 and SAT2 capsid protein (Figure 2C), Lpro, and P2 and P3 genomes regions of FMDV (Appendix A). The trees were constructed using the sequences obtained by plaque assay, first passage, and references. With respect to SAT1 sequences, #51/SAT1 plaque sequences are similar to the assembly obtained from the co-infected first-passage isolate. However, the first-passage sequences from #6/SAT1, #61/SAT1, and #36/SAT1 may have been incorrectly called due to co-infection based on their divergence from the plaque-isolated sequences (Figure 2). With respect to SAT2, the sequences obtained from first-passage samples were similar to the ones obtained by plaque assay; however, the two distinct genotypes obtained from animals #36 and #59 (SAT2 A and B) were not resolved by analysis of the first-passage isolate (Appendix A).

### 3.3. Recombination Detection and Bayesian Time Divergence Estimates

Six recombination breakpoints were identified in the alignment with significant *p*-values across the FMDV genome; specifically, two in VP4, two in 2B, one in 2C, and one in 3C. Based on the results of the breakpoint analysis, Bayesian phylogenies were reconstructed using the recombination-free coding regions: 1-633 (Leader Proteinase-Lab), 766-2889 (VP4 partial, VP2, VP3, VP1), 3148-4235 (2B partial/2C partial 3′), 4236-5295 (3A/3B/3C partial 3′), 5295-7029 (3C partial 5′/3D) (Figure 3). The incongruent topologies of each of the segments supported the results of the breakpoint analysis. First-passage sequences from co-infected samples were excluded due to potential assembly artifacts.

#### 3.3.1. Capsid-Coding Segment (ORF Alignment Positions 766-2889)

In the reconstruction of the capsid-coding segment, sequences from animal #6/SAT1 diverged earliest from the #36/SAT1 and #51/SAT1 sequences, with a most recent common ancestor estimated in 1987 (95% HPD 1971–2001; Figure 3). Animal #36/SAT1 and #51/SAT1 sequences were more closely related (TMRCA: 2001 95% HPD 1992–2009). The estimated ancestor for all sequences of #51/SAT1 was 2014 (95% HPD 2012–2015), indicating a higher within-host divergence of this virus. SAT2 capsid sequences from #36/SAT2, #51/SAT2, and #6/SAT2 were closely related, with an estimated TMRCA in 2011 (95% HPD 2008–2014). Interestingly, there were two different clusters of SAT2 viruses from animal #36. These two SAT2 viruses were denoted as #36/SAT2A and #36/SAT2B. The sequences had a recent estimated common ancestor in 2015 (95% HPD 2014–2016) based on the capsid sequences. Notably, animal #59, which was initially identified as co-infected, but from which only SAT2 was recovered from plaques, also had two different SAT2 viruses, denoted #59/SAT2A and #59/SAT2B. These viruses had a recent common ancestor estimated at 2014 (95% CI 2013–2015) based on the capsid sequences. Interestingly, one virus within the #59/SAT2A group (named #59/SAT2-plaque4) was considerably divergent (marked with an * in Figure 3) from both #59SAT2A and #59SAT2B clades. This virus had a unique phylogeny not only in the capsid-coding region, but throughout all recombination-free coding areas.

#### 3.3.2. Leader Proteinase Coding Segment (ORF Alignment Positions 1-633)

The topology of Lpro grouped predominantly by serotype, except for #36/SAT2A, which grouped with the SAT1 viruses (Figure 3). The TMRCA estimated for SAT1 and SAT2 was within the 95% HPD of the capsid segment estimates. For example, the TMRCA for all SAT1 sequences was estimated at 1987 (95% HPD 1971–2001) by the capsid phylogeny, and 2001 (95% HPD 1995–2007) by the leader proteinase segment. Similarly, estimation of TMRCA for all SAT2 (excluding divergent #36/SAT2A) was 2005 (95% HPD 1999–2010) using leader proteinase, whereas it was 2011 (95% HPD 2008–2014) for the capsid segment. The genetic diversity within animals was low (mean nucleotide difference = 0 for all groups, except #51/SAT1, which had a mean nucleotide difference of 2.6). #59/SAT2-plaque4 chimeric virus had a mean difference of eight nucleotides with #59SAT2A, and seven with #SAT2B. There was a remarkable divergence between the #36/SAT2A and #36/SAT2B; the leader proteinase TMRCA was estimated at 1982 (95% HPD 1970–1993) in contrast with the capsid-coding region TMRCA estimated at 2015 (95% HPD 2014–2015).

#### 3.3.3. 2B Partial/2C Partial (ORF Alignment Positions 3148-4235)

The topology in the 2B partial/2C partial segment differed from that of the capsid and Lpro segments (Figure 3). Interestingly, in this segment, viruses from the same animal tended to share a close genetic relationship regardless of serotype. Specifically, viruses #51/SAT1 and #51/SAT2 were grouped together, as were #6/SAT1 and #6SAT2, and#36/SAT1 and#36/SAT2A. The single exception was #36/SAT2B, which grouped with sequences distant from #36/SAT1 and #36/SAT2A. TMRCA was estimated at 2008 (95% HPD 2002–2011), 2013 (95% HPD 2011–2015), and 2010 (95% HPD 2007–2013) for SAT1/SAT2 sequences from animals #51, #6, and #36 (excluding #36/SAT2B), respectively. The TMRCA for animal #36 sequences including the divergent group was 1981 (95% HPD 1967, 1992). Similar to the other segments, the within-group nucleotide difference was greater for animal #51/SAT1 (mean nucleotide difference = 9.6) compared to other groups (mean differences #36/SAT1 = 0.3, #51/SAT2 = 0.2, #6/SAT1 = 0.2, 59SAT2B = 0.5, and 0 for the remaining groups). The chimeric virus #59/SAT2-plaque4 had a mean nucleotide difference of 13 with #59/SAT2A, and 14.25 with #59/SAT2B.

#### 3.3.4. 3A/3B/3C Partial (ORF Alignment Positions 4236-5295)

The topology of 3A/3B/3C partial was similar to the 2B/2C topology, with viruses grouping by host animal rather than serotype. This trend was upheld even in the divergent groups of animal #36 (SAT1 and SAT2A and B). The TMRCA between the SAT1 and SAT2 viruses for animals #51, #6, and #36 was 2011 (95% HPD 2008–2014), 2010 (95% HPD 2005–2014), and 2007 (95% HPD 2002–2011), respectively. The branching pattern of #51/SAT1 had three divergent sequences. Within this genotype, three subclades are distinguishable in most regions with a common ancestor. Although these sequences appear more closely related to the SAT2 viruses from the same animal, the small number of nucleotide differences between them may not provide a conclusive resolution of the clades. The chimeric virus #59/SAT2-plaque4 had a mean nucleotide difference of 9 with #59/SAT2A, and 20 with #59/SAT2B.

#### 3.3.5. 3C Partial/3D (ORF Alignment Positions 5296-7029)

The topology of the coding sequences for 3C-3D was similar to 2B partial/2C partial and 3A/3B/3C partial topologies (Figure 3). The TMRCA for SAT1 and SAT2 sequences collected from animal #51 was 1997 (95% HPD 1985–2010), which is 14 years earlier than the estimation by the 3A/3B/3C partial and 2B partial/2C partial phylogenies. The TMRCA for #36/SAT1 and #36/SAT2 was 2010 (95% HPD 2006–2013), and the TMRCA for #6/SAT1 and #6/SAT2 was 2009 (95% HPD 2002–2013), similar to the estimates of the 3A/3B/3C partial and 2B partial/2C partial phylogenies. The chimeric virus #59/SAT2-plaque4 had a mean nucleotide difference of 14.06 with #59/SAT2A, and 11.00 with #59/SAT2B groups.

### 3.4. Within-Host Recombination Network Analysis

The phylogenetic analyses indicated that one plaque-derived virus (#59/SAT2-plaque4) had a unique branching pattern. The genetic relationship of this virus was not clearly associated with a particular genotype. A network analysis using all plaque-derived sequences from animal #59 was generated to investigate #59/SAT2-plaque4 as a potential chimera of #59/SAT2A and #59/SAT2B (Figure 4A).

Network analysis placed this sequence in an intermediate path between #59/SAT2A and #59/SAT2B, which may indicate an evolutionary potential recombinant (chimeric) virus (Figure 4). To further rule out assembly problems related to low coverage, low quality/short reads, or partial read alignment, #59/SAT2-plaque4 was queried on multiple platforms to ensure sequence quality. The #59/SAT2-plaque4 raw reads, when mapped to the SAT2/KEN 002/2002 reference (Genbank #JF749861), had an average read length of 144.73, minimum coverage of 344, and maximum coverage of 2850.

Network analysis was also conducted for #51/SAT1 and #51/SAT2 due to the significantly higher genetic diversity of #51/SAT1 to exclude the existence of a chimeric virus (Figure 4B). Interestingly, the diversity of #51/SAT1 was similarly divergent in three subgroups.

### 3.5. Within-Host FMDV Genotype Sequence Diversity

Ten distinct representative genotypes were identified according to the topology of the phylogenetic trees (Figure 3). Animal #36 had three different viruses: #36/SAT2A, #36/SAT2B, and #36/SAT1. The nucleotide identity between #36/SAT2A and #36/SAT2B P1 regions was 98.3%, whereas the amino acid identity was 98.8%. Animals #6 and #51 had one SAT1 and one SAT2 each. Animal #59 had two SAT2 viruses, and #61 had one SAT2 virus (Table 1). Of all the viruses sequenced, #36/SAT1 (*n* = 7) and #36/SAT2A (*n* = 6) were completely clonal. Sequences from #6/SAT1, #6/SAT2, #36/SAT2B, #51/SAT1, #51/SAT2, #59/SAT2A (excluding the chimeric sequence #59/SAT2-plaque4), #59/SAT2B, and #61/SAT2 contained polymorphic sites, varying between 1–5 SNP sites per group.

Virus #51/SAT1 contained a high number of polymorphic sites throughout the entire genome (Figure 5A). Out of 7008 total sites, 76 (1.08%) were polymorphic in #51/SAT1. Notably, 66 polymorphic sites in #59 SAT2A were due to a single sequence (plaque4). The remaining six viruses had between 1–5 polymorphic sites across the genome.

Transitions and non-synonymous mutations were frequent throughout the genome of the eight viruses. The majority of nucleotide substitutions were C → T (30.4%) and T → C (25%), followed by A → G (15.2%) and G → A (12%) (Figure 5B). No trends in the location of nucleotide substitutions across the genome were observed. Interestingly, non-synonymous mutations were located throughout the genome in the highly variable virus #51/SAT1, except in the 2C protein-coding region. However, non-synonymous mutations were located in the 2C region in less variable viruses #61/SAT1 and #51/SAT2.

## 4. Discussion

African buffalo are the main wildlife reservoir of FMDV in sub-Saharan Africa. FMDV can circulate subclinically in buffalo for up to 5 years [35], and individual animals can harbor multiple serotypes [12]. However, little is known about the viral dynamics and evolution of FMDV during naturally occurring co-infections. In the current study, we used plaque purification coupled with deep sequence analysis to increase the resolution of unique FMDV sequences obtained from buffalo co-infected with SAT1 and SAT2. In some cases, the viral diversity characterized in the co-infected animals revealed the simultaneous presence of related genotypes within a serotype [36]. Additionally, phylogenetic grouping of non-structural protein-coding sequences of the viruses sampled from the same animals tended to have a close genetic relationship, suggesting that interserotypic recombination had occurred within those animals.

In addition to the SAT1-SAT2 co-infections recovered from the plaque assays in animals #36, #51, and #6, we identified two SAT2 variant viruses in animals #59 and #36. Within each animal, there was a small but clear divergence in the capsid sequences, and the SAT2 variants were very distant in other regions of the genome, such as Lpro (#36). This is the first time that naturally occurring co-infection of different genotypes from the same serotype have been reported resulting in a nucleotide identity of 98.3%. The presence of such a diversity of viruses within a single host, and the inconsistencies of tree topologies between different coding areas of the virus, suggest that recombination occurs often, perhaps within these animals, as seen for animal #36/SAT2A and #36SAT2B (TMRCA = 2010 and 2011 in 3A-3D). This is consistent with previous reports of recombination within the FMDV genome [25,37,38]. A recent experimental study in cattle demonstrated that serotype A/O recombination occurred in the nasopharynx of serotype A-infected carriers within 10 days of superinfection with FMDV-O [39].

Phylogenetic analyses revealed an expected discrepancy of the tree topologies for distinct regions of the virus (Lpro, 2B-2C, 3C-3D). The six recombination breakpoints identified for SAT1 and SAT2 in the current study (two in VP4, two in 2B, one in 2C, and one in 3C) are similar to those identified previously for serotype O and A viruses [25,40]. The topology discrepancies in non-structural regions are conserved within-host, even between SAT1 and SAT2 serotypes, suggesting significant host-derived selection occurs in non-structural coding regions following FMDV infection.

As a host species with long-term asymptomatic infection, buffalo can be co-infected with multiple FMDV genotypes, creating the conditions necessary for the emergence of new variants associated with within-host evolution and recombination. For example, the #59/SAT2-plaque4 sequence represents a potentially unique combination of #59/SAT2A and #59/SAT2B plaque sequence groups. Further assessing the quality of the assembly robustly supported the accuracy of the within-plaque consensus sequence. Although we cannot determine whether the chimeric sequence was a result of recombination that occurred within the animal or during sample preparation and analysis, this observation is relevant to understand within-host viral evolution. Bioinformatic errors, particularly the assembly of short reads for samples with viral co-infections, complicate the differentiation of multiple similar sequences from the same dataset (Figure 2A,B). In this study, it is possible that plaque purification resulted in plaques seeded by multiple virus particles, thereby confounding some results. However, in animal #36/SAT2 and #59/SAT2 plaques, both groups A and B were made up of at least four identical viral genomes, suggesting improper plaque purification did not take place, and bioinformatic analyses did not confound the results. It is unlikely that each of the four plaques were seeded by two different, but identical, virus particles to produce these results, further confirming the results in this study.

The potential for, and prevalence of, FMDV recombination have received substantial attention recently [20,38,39]. Though the mechanism of FMDV recombination remains incompletely elucidated, it is thought to occur by a copy-choice recombination, which requires co-infection of a single cell with two distinct viruses [41]. This mechanism has been identified for other picornaviruses, such as enteroviruses and rhinoviruses [42]. Widespread persistent and neoteric subclinical infection of African buffalo with multiple FMD viruses provides an opportunity for copy-choice recombination, and likely contributes to FMDV strain diversity and emergence. Multiple studies have found statistically significant evidence of FMDV recombination based on historical FMDV sequences [25,38].

The time to most recent common ancestor (TMRCA) estimated for each recombination-free region can be used to infer past recombination events. For example, #36/SAT2 A and B capsid region sequences had an estimated TMRCA of 2015, whereas the remaining partitioned genomic regions had an earlier TMCRA estimate (1989–2012). The divergence of Lpro and P2 suggest different evolutionary origins of these two viruses. Considering the long duration of FMDV infections in African buffalo [35], and the age of this buffalo (5 years), it is possible that a recombination event from different viruses occurred in the animal. It is possible that viruses with the observed #36SAT2 capsid out-competed the parental virus that contributed one of the divergent non-structural protein clades. The divergent grouping of the two related capsid clades (#36/SAT2A and #36SAT2B) may have been driven by the differences in non-protein-coding regions.

In this study, we assessed both the plaque-purified sequences, as well as the first-passage sequences obtained from co-infecting viruses. Our analyses suggested that it would be difficult to separate the sequence data from the first passage into the unique viral groups that were identified in the plaque-derived sequences. Of the 10 genotypes identified by plaque purification in 5 animals, one virus, #51/SAT1, had a high within-animal genetic diversity. Specific genotype and cluster information could only be resolved with plaque purification, whereas without plaque purification, the broad viral diversity in these samples would not allow for proper genome assembly. Although plaque purification will selectively filter out certain viruses, it represents a unique modality to study subpopulations of viruses in the host. Our results also suggest that sequences from co-infected samples obtained using NGS without plaque purification may be inaccurate, and should be examined critically when used in downstream analyses. It is highly problematic that, in many cases, coinfection would be missed, leading to misinterpretation of sequence data and epidemiological findings.

FMDV often circulates subclinically in the buffalo population, and the virus is extremely successful in persisting in the population. Studies in several geographic regions have reported a high seroprevalence, indicating exposure to multiple serotypes in buffalo [13,15]. The mechanisms of FMDV persistence within the buffalo population has been an important area of investigation because most studies fail to observe transmission from persistently infected animals. A recent study combining experimental infection and modeling concluded that this persistence cannot be sustained only by infections in young calves after waning of maternal immunity, and that virus transmission from persistently infected carrier animals is needed for maintenance of the virus in the population [43]. Animals persistently infected with multiple FMDV genotypes may play an important role in virus evolution and divergence, and in the antigenic differences, allowing for infection or transmission to a new host. This viral diversity is also important due to the potential emergence of variants that are more fit and likely to cause occasional ‘spill-over’ to livestock [7,44,45]. Although the diversity of FMDVs in the study animals is clear, the epidemiological significance cannot be addressed by the retrospective, descriptive study design of the current investigation. It is not clear if the samples in this study were collected from persistently or acute subclinically infected animals. Additionally, it is unclear if these viruses were associated with clinical disease in these or any other animals. Future directions should aim at a longitudinal sampling of persistently infected buffalo that may provide essential genetic information at different times during infection. Utilizing these samples to isolate viruses would provide evidence of genetic and phenotypic differences that confer the ability to replicate, persist, and transmit to non-reservoir hosts.

Widely used methods of short-read NGS have triggered abundant genomic research that has deepened our understanding of viruses’ evolution. However, this tool has an important limitation when studying within-host viral diversity, due to the inability to correctly assemble closely related viral genomes. In this study, we applied plaque purification coupled to deep sequencing to investigate FMDV viral population dynamics in co-infected buffalo. This method confirmed the presence of recombination in co-infected samples with a higher level of confidence and resolution than would otherwise be possible. Additionally, plaque purification revealed that subclinically co-infected animals maintained up to three distinct viruses simultaneously. These findings improve our understanding of how the diversity of FMDVs co-existing within a subclinically infected host may provide optimal conditions for viral recombination and the emergence of novel viral variants. Further investigation may elucidate whether such processes lead to altered fitness, transmissibility, or spill-over to susceptible livestock or other wildlife hosts.

## Figures and Tables

**Figure 1 viruses-14-00897-f001:**
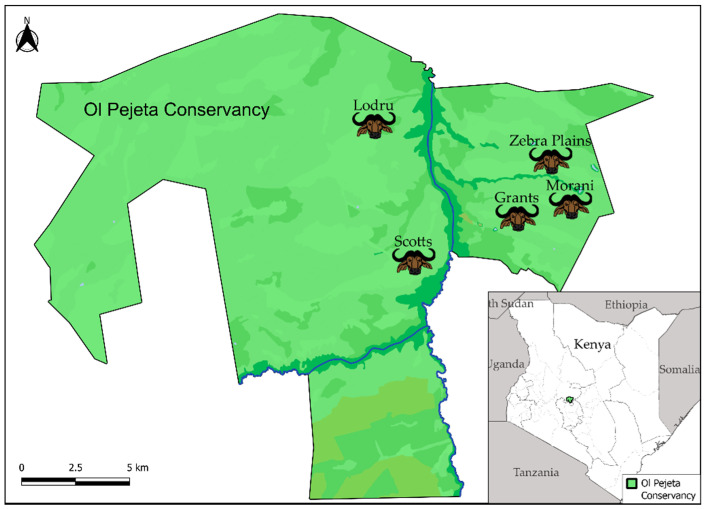
Buffalo herd geographic distribution in the Ol Pejeta Conservancy (OPC). A map of the OPC indicating the location of each sampled buffalo herd (black text indicates the name, the buffalo head indicates the location in the conservancy where the animals were sampled). The Ewaso Ngiro River is indicated by a blue line. The specific location of the OPC in central Kenya appears in the inset on the bottom right.

**Figure 2 viruses-14-00897-f002:**
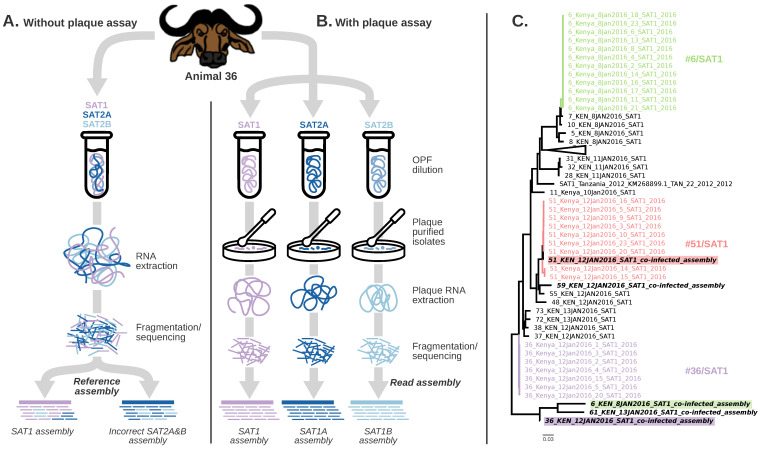
Assembly error induced by viral co-infections. (**A**) Describes the assembly process without and (**B**) with plaque purification. (**C**) Maximum likelihood phylogeny of the capsid protein including SAT1 sequences from co-infected samples obtained by direct NGS from the first passage of the sample, and from individual plaque-purified viral. The highlighted sequences depict an incorrect assembly of consensus SAT1 sequences obtained from first-passage co-infected samples. Short reads of related viruses resulted in a biased consensus. Therefore, these sequences have an apparent distant genetic relationship from the sequences obtained with the plaque purification assay.

**Figure 3 viruses-14-00897-f003:**
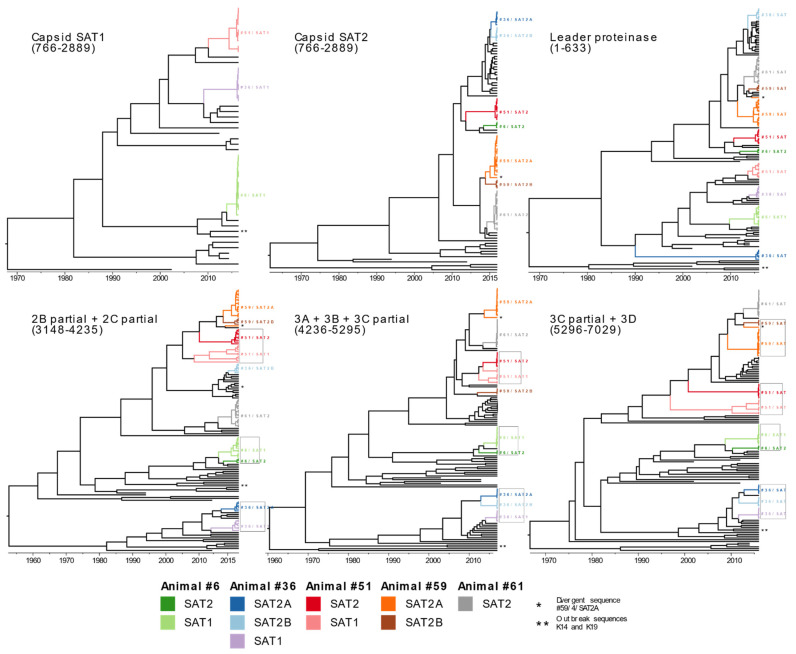
Maximum clade credibility tree of recombination-free regions. The color of the sequence groups follows the grouping based on the capsid region (nucleotide positions 766-2889) encompassing the coding genes of VP2, VP3, and VP1. Six different genotypes were identified in the co-infected animals. The (*) indicates a divergent sequence from the #59/SAT2A that diverges from its main group throughout all phylogenies. The (**) indicates the location of the outbreak sequences K14 and K29.

**Figure 4 viruses-14-00897-f004:**
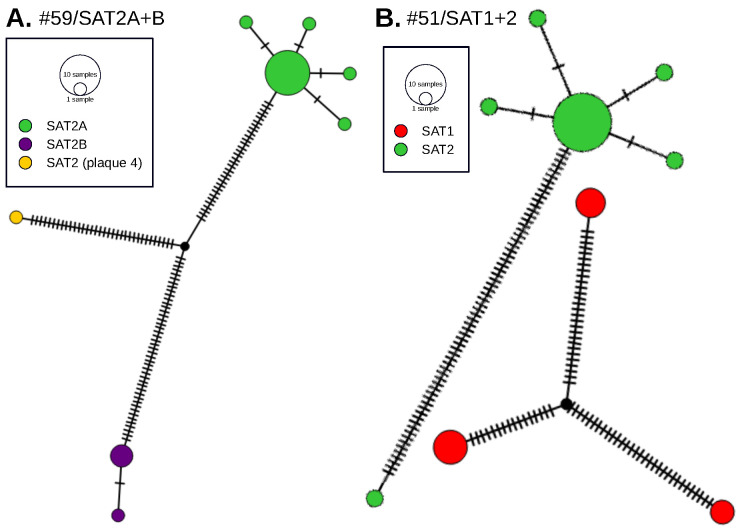
Network analysis of plaque sequences. (**A**) One sequence from this animal (#59/SAT2-plaque4) is a potential chimera of #59/SAT2A and #59/SAT2B. The network analysis confirms this observation by creating a network in which two genotypes originate from the same node. (**B**) Representative datasets were used from #51/SAT1 and #51/SAT2. Despite the divergence of the sequences, there is no apparent shared network between the SAT1 and SAT2 viruses.

**Figure 5 viruses-14-00897-f005:**
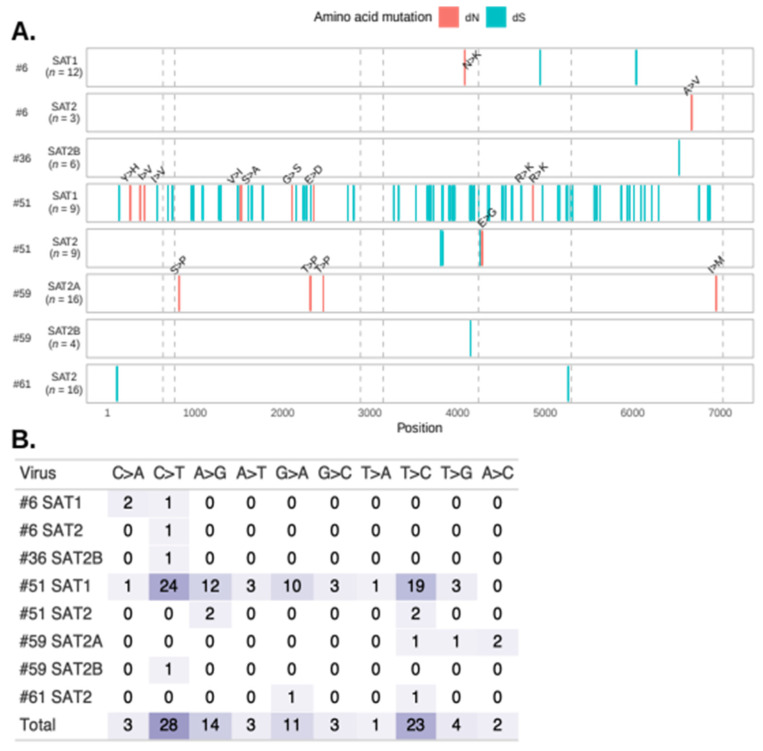
Location and frequency of single nucleotide polymorphisms in FMDV plaque-purified samples. (**A**) Location of synonymous (dS; teal) and nonsynonymous (dN; red) mutations across each viral subtype per animal. Nonsynonymous mutations are annotated, where the less frequent amino acid is treated as the mutated amino acid. Chimeric sequence #59/SAT2-plaque4 was excluded. The grey dashed lines indicate the recombination breakpoints identified by RDP4 (Appendix A). The total number of sequences (*n*) for each alignment are reported. (**B**) Frequency of specific nucleotide polymorphisms in the plaque sequences.

**Table 1 viruses-14-00897-t001:** Isolates obtained from plaque purification of oropharyngeal fluid of African buffalo. The supernatants of the virus isolated from the original clinical sample were serially diluted and plaque-purified in LFBK-αvβ6 cells. The resulting supernatants were subjected to whole genome sequencing and genotypic analysis.

Buffalo ID	6		36			51		59		61		Total
Dilution of OPF	10^−2 > −3^	10^−4 > −6^	10^−2 > −6^			10^−1 > −2^	10^−3 > −6^	10^−2 > −3^	10^−4 > −6^	10^−1 > −2^	10^−3 > −6^	-
# Purified Plaques	14	8	22			21	2	14	8	22	1	112
Plaque Serotypes	SAT1	SAT2	SAT1	SAT2		SAT1	SAT2	SAT2		SAT2		-
# Full CDs	12	3	7	12		9	9	21		16		89
Genotype	-	-	-	A	B	-	-	A	B	-		-
# Isolates Obtained	12	3	7	6	6	9	9	17	4	16		89
# Sequences Analyzed	12	3	7	12		9	9	21		16		89

Abbreviations: Virus isolate (VI); Coding domains (CDs); Number of (#).

## Data Availability

FMDV genomes are available through Genbank under accessions OM562453-OM562541 and OM562542-OM562602. This manuscript is referring to the first version.

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
