# Peer review of "Viral Population Diversity during Co-Infection of Foot-And-Mouth Disease Virus Serotypes SAT1 and SAT2 in African Buffalo in Kenya"

_viruses, 2022, doi:10.3390/v14050897_

Round 1

Reviewer 1 Report

In this manuscript, the authors investigated viral population diversity during co-infection of FMDV SAT1 and SAT2 in African Buffalo and demonstrated that viral evolution through recombination and point mutation contributes to broad viral diversity, which may be critical to FMDV persistence in the individual animal. The manuscript was prepared well although there are still a few of errors in this manuscript and careful editing should be done. 

1 Authors concluded that the diversity of FMDVs co-exists within a sub-clinically infected host which may provide optimal conditions for viral recombination and emergence of novel viral variants. Do authors have any evidence that an FMD outbreak was caused by such a novel viral variant derived from the viral recombination and point mutation reported in this study, which authors need to

2 Authors found that the interserotypic recombination had occurred within those animalsco-infected with FMDV SAT1 and SAT2, but there is lack of biological function research. Did authors find such viral variant derived from interserotypic recombination in the OPF samples from acutely infected animals in the same region?

Minor Issues:

Line 106:OPF should be corrected to Oropharyngeal fluid (OPF)

Line 109:Oropharyngeal fluid (OPF) should be corrected to OPF

Line 109: what’s buffer used for collecting OPF?

Figure 3: The quality of words in figure needs to be improved.

Author Response

The authors would like to thank the reviewer for taking the time to provide constructive comments. We have responded to each comment in blue below.

In this manuscript, the authors investigated viral population diversity during co-infection of FMDV SAT1 and SAT2 in African Buffalo and demonstrated that viral evolution through recombination and point mutation contributes to broad viral diversity, which may be critical to FMDV persistence in the individual animal. The manuscript was prepared well although there are still a few of errors in this manuscript and careful editing should be done.

  1. Authors concluded that the diversity of FMDVs co-exists within a sub-clinically infected host which may provide optimal conditions for viral recombination and emergence of novel viral variants. Do authors have any evidence that an FMD outbreak was caused by such a novel viral variant derived from the viral recombination and point mutation reported in this study, which authors need to

Thank you for the comment. It is not clear if an outbreak can be attributed to the viruses described in the study. We have added the following information to lines 503-508 to address the suggested point, “Although the diversity of FMDVs in these animals is clear, the epidemiological significance cannot be addressed by the retrospective, descriptive study design of the current investigation. It is not known if the samples we have described were derived during the persistent or acute subclinical phase in these animals. Further, it is no known if any of these viruses were associated with clinical disease in these, or any other animals.”

  1. Authors found that the interserotypic recombination had occurred within those animalsco-infected with FMDV SAT1 and SAT2, but there is lack of biological function research. Did authors find such viral variant derived from interserotypic recombination in the OPF samples from acutely infected animals in the same region?

Thank you for the constructive comment. The manuscript by Ferretti et al 2018 entitled, “Within-Host recombination in the Foot-and-Mouth Disease Virus genome,” describes interserotypic recombination within a set of buffalo samples sourced from the same region. It is not clear if the buffalo were acutely infected or persistently infected as African buffalo demonstrate minimal to no clinical FMDV upon infection and are not monitored in a manner that could provide the suggested information.

Minor Issues:

Line 106:OPF should be corrected to Oropharyngeal fluid (OPF)

Thank you for the comment. The correction was made in the text.

Line 109:Oropharyngeal fluid (OPF) should be corrected to OPF

Thank you for the comment. The correction was made in the text.

Line 109: what’s buffer used for collecting OPF?

Thank you for the comment. No buffer is used, the probing cup is utilized to collect fluid from the animals without buffer. The animals produce enough fluid that the utilization of additional fluid is not needed.

Figure 3: The quality of words in figure needs to be improved.

Thank you for the comment. The figure has been provided to the journal in a higher resolution .tiff file for publication.

Reviewer 2 Report

The study by Palinski et al. is interesting  and this study provides novel information about the circulation of these viral agents in natural reservoirs of FMDV in sub-Saharian Africa

The study is well performed and presented clearly

Author Response

The authors thank the reviewer for taking the time to review our manuscript and appreciate the feedback.